# Climate and ecology predict latitudinal trends in sexual selection inferred from avian mating systems

**Robert A. Barber** [1] *, **Jingyi Yang** [1], **Chenyue Yang** [1], **Oonagh Barker** [1,2], **Tim Janicke** [3], **Joseph A. Tobias** [1] *

**1** Department of Life Sciences, Imperial College London, Ascot, United Kingdom, **2** School of Biological Sciences, University of East Anglia, Norwich, United Kingdom, **3** CEFE, Université de Montpellier, CNRS, EPHE, IRD, Montpellier, France

* r.barber19@imperial.ac.uk (RAB); j.tobias@imperial.ac.uk (JAT)

**Data Availability Statement:** Data and code used in this study are available at the cited sources, and at https://doi.org/10.6084/m9.figshare.27255609. Species-level data for all analyses and figures are

## Abstract

Sexual selection, one of the central pillars of evolutionary theory, has powerful effects on organismal morphology, behaviour, and population dynamics. However, current knowledge about geographical variation in this evolutionary mechanism and its underlying drivers remains highly incomplete, in part because standardised data on the strength of sexual selection is sparse even for well-studied organisms. Here, we use information on mating systems—including the incidence of polygamy and extra-pair paternity—to estimate the intensity of sexual selection in 10,671 (>99.9%) bird species distributed worldwide. We show that avian sexual selection varies latitudinally, peaking at higher latitudes, although the gradient is reversed in the world's most sexually selected birds—specialist frugivores—which are strongly associated with tropical forests. Phylogenetic models further reveal that the strength of sexual selection is explained by temperature seasonality coupled with a suite of climate-associated factors, including migration, diet, and territoriality. Overall, these analyses suggest that climatic conditions leading to short, intense breeding seasons, or highly abundant and patchy food resources, increase the potential for polygamy in birds, driving latitudinal gradients in sexual selection. Our findings help to resolve longstanding debates about spatial variation in evolutionary mechanisms linked to reproductive biology and also provide a comprehensive species-level data set for further studies of selection and phenotypic evolution in the context of global climatic change.

## Introduction

Many of the most spectacular outcomes of animal evolution, from the peacock's tail to the song of the nightingale, are testament to the power of sexual selection, a pervasive mechanism driven by competition among individuals for reproductive success [1]. Decades of intensive research have shown that sexual selection has far-reaching impacts on adaptation [2], speciation [3], and population dynamics [4], with implications for genomic evolution [5] and the

presented in S1 Data and S2 Data; all R code for analyses and figures is available at https://www.github.com/ra-barber/sexual_selection.

**Funding:** This study was funded by a Natural Environment Research Council PhD scholarship to RAB, hosted by the Quantitative Methods for Ecology and Evolution Centre for Doctoral Training, Imperial College London (Doctoral Training Centre: NE/P012345/1). The funders had no role in study design, data collection and analysis, decision to publish, or preparation of the manuscript.

**Competing interests:** The authors have declared that no competing interests exist.

**Abbreviations:** EPP, extra-pair paternity; OSR, operational sex ratio; OSS, opportunity for sexual selection; SAR, simultaneous autoregression; VIF, variance inflation factor.

response of organisms to environmental change [6–8]. Previous studies focusing on local contexts or experimental systems have made substantial progress in understanding how sexual selection varies within populations [9–11], yet the factors explaining variation among species remain unclear, particularly at wider taxonomic and geographic scales [12–15].

Large-scale patterns in reproductive strategies have often been proposed, with one recurring suggestion being that sexual selection may vary with latitude or climate. However, both the existence and the direction of these potential gradients remain disputed (Fig 1). Some studies suggest that the intensity of sexual selection is likely to increase at higher latitudes where shorter reproductive periods and higher breeding synchrony intensify competition for mates [16,17], while also presenting more opportunities for polygyny [18] and extra-pair matings [19]. This latitudinal pattern could be accentuated by widespread year-round defence of large pair- or group-territories in tropical animals, which may constrain population density, thereby reducing prospects for polygamy [20]. Conversely, other studies suggest that sexual selection will peak in the tropics, either because year-round reproduction and asynchronous breeding allows males to mate with multiple females sequentially [21] or because long fruiting or flowering seasons can promote either resource defence polygyny or female-only parental care [15], leading to extreme forms of male–male competition, such as lekking [22,23].

These alternative hypotheses are difficult to disentangle and previous tests have produced conflicting results. Specifically, some analyses suggest that sexual selection is most intense at higher latitudes [17,24,25], whereas others find evidence that sexual selection either increases in the tropics [23,26] or has no significant relationship with latitude in either direction [27–30]. A major limitation is that numerous attempts to examine macroecological patterns in reproductive strategies have relied on secondary sexual traits, such as sexual size dimorphism or sexual dichromatism, as indices of sexual selection [17,25,26,31–34]. These metrics can only provide an incomplete picture of variation in reproductive strategies because they overlook sexual selection acting on different traits, such as acoustic signals [35]. The predictive power of standard morphological metrics for sexual selection are also weakened by mutual sexual selection [36] and social selection [37], both of which can drive increased body size and ornamentation in females [38]. A more complete test of sexual selection gradients and their drivers

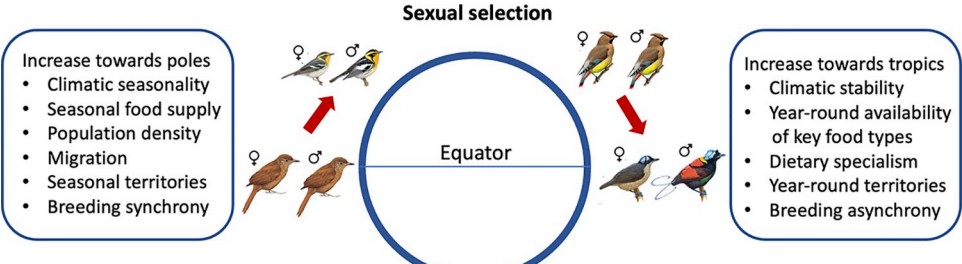

**Fig 1. Potential latitudinal gradients in sexual selection and their putative mechanisms.** One hypothesis proposes that avian sexual selection increases with latitude because highly seasonal climates promote short, synchronous breeding seasons, leading to intense competition for mating opportunities in dense populations. This concept may apply primarily to higher trophic levels, with a gradient from monomorphic insectivores with year-round territories in the tropics (for example, Uniform Treehunter) to highly dimorphic migratory insectivores at higher latitudes (for example, Blackburnian Warbler). An alternative hypothesis predicts that sexual selection increases towards the tropics because climatic stability promotes year-round availability of food resources, including abundant fruit and flowers, leading to extreme polygamy in dietary specialists. This concept may apply primarily to lower trophic levels, with a gradient from monomorphic frugivores with biparental care at higher latitudes (for example, Japanese Waxwing) to highly dimorphic frugivores with female-only care in the tropics (for example, Wilson's Bird-of-Paradise). Images are from Birds of the World, reproduced with permission of Cornell Lab of Ornithology.

requires direct evidence from behavioural observations and parentage analyses, yet this information has not previously been available at the required taxonomic and geographic scale.

To provide a new perspective on global patterns, we quantified variation in the strength of sexual selection inferred from mating systems and associated behaviours across 10,671 (>99.9%) bird species worldwide. We compiled standardised scores of sexual selection using a well-established protocol based on the degree of polygamy reported in each species [32,39–41] with modifications based on information about extra-pair paternity (EPP). Although mating systems do not provide a complete measure of sexual selection [42], they can serve as a robust and objective proxy [43–45] with fewer limitations than widely used morphological indices [35]. Clearly, more direct measures of sexual selection would be preferable, but such measures are currently only available for relatively few species, and therefore not suitable for exploring global gradients.

Birds currently offer the best available system for a global synthesis of sexual selection because we know more about their breeding behaviour than any other major taxonomic group [46–48]. In particular, a growing number of studies report information on relatively cryptic forms of sexual selection, including molecular analyses of parentage revealing the degree of EPP [21,47,49]. Furthermore, information on breeding can be coupled with uniquely comprehensive data sets on avian phylogenetic relationships, geographical distribution, life history, and ecology [50–52]. By combining these resources for all birds, along with spatial analyses and phylogenetic models, we test key hypotheses about fundamental drivers of macroevolutionary patterns in sexual selection [12,13].

We begin by assessing whether sexual selection scores are correlated with the latitude at which species occur, and then—because latitudinal gradients are best viewed as emergent properties shaped by multiple underlying mechanisms [15]—we evaluate the role of climate and ecology as fundamental drivers of variation in sexual selection across species. Specifically, we conduct a range of analyses to assess the relative importance of diet [22,53], migratory behaviour [54], territoriality [55], and climatic seasonality [56], all of which have been proposed to regulate the intensity of sexual selection and thus to explain the enormous variation in sexual traits observed worldwide [15,20,23]. Most hypotheses predict that sexual selection is accentuated by abundant food supply, migration, smaller home ranges, or seasonal environments, but the relevance of these factors at macroecological scales remains unknown.

## Results

We scored sexual selection by assigning 10,671 bird species (eBird taxonomy [57]), including all 9,988 valid bird species included in the global BirdTree phylogeny [50], to one of 5 categories ranging from 0 (strict monogamy) to 4 (extreme polygamy). These scores were assigned according to thresholds based on the degree of polygamy and EPP (Table 1; see Methods). Focusing on the BirdTree dataset, higher sexual selection scores (1–4) were assigned to 1,750 species (18%), while most species (*n* = 8,238; 82%) were scored as strictly monogamous (S1a Fig).

The predominance of strict monogamy may partly reflect a tendency to underestimate sexual selection in the literature. For example, species observed in socially monogamous pairs are generally assumed to be genetically monogamous, whereas further study often reveals them to have higher rates of polygamy and EPP [49]. Given the potential for poorly known species to be misrepresented in our data set, we also scored each species based on the quality of data available (Table A in S2 Text). When we focused exclusively on the best-known species with highest certainty data (*n* = 2,793), the proportion of strictly monogamous species was lower (*n* = 1,826, 65%), while a third of species (*n* = 967, 35%) were classified as either polygamous

**Table 1. Criteria for scoring the intensity of sexual selection in birds.** All species were scored from literature by applying this classification system to textual descriptions of mating systems and breeding behaviour. Where available, we used quantitative estimates of polygamy and EPP to refine our scores. The thresholds for scores 0–3 were based on previous studies [39–41]. We treated lekking as an extreme form of polygyny based on genetic evidence of very high variance in mating success in most well-studied cases (see Methods; Fig 2). Where scores differed among criteria, we selected the highest score. Scores for all species are presented in S1 Data along with literature sources and estimates of data certainty.

| Score | Criteria |
|---|---|
| 0 | Strict monogamy, including extreme low rates (<0.1%) of individual polygamy and <5% extra-pair paternity offspring (EPP). |
| 1 | Frequent monogamy, including relatively low rates (0.1%–5%) of individual polygamy and <25% EPP. Species with occasional records of polygamy were assigned this score. |
| 2 | Regular polygamy, including cases with multiple records of polygamy. Where empirical data are available, this score was applied to species with 5%–20% of individuals polygamous. This score was also applied to species with 25%–50% EPP. |
| 3 | Frequent polygamy with >20% of individuals polygamous or very high rates of EPP (>50% of offspring). |
| 4 | Extreme polygamy characterized by communal display (lekking) behaviour or permanent display posts used by solitary individuals. This score was not assigned according to levels of polygamy or EPP. |

or subject to sexual selection via EPP (scores 1–4). Sexual selection and data-certainty scores for all bird species, aligned with both BirdTree and eBird taxonomies, are presented along with sources in S1 Data.

To assess the validity of our sexual selection scores, we compared them against 3 widely used metrics of sexual selection: residual testes mass as an index of post-copulatory sperm competition [58], Bateman gradients reflecting the increase in reproductive success obtained from additional matings [59], and opportunity for sexual selection ($I_S$) reflecting variance in mating success [60] (for a full explanation, sources and derivation of these metrics, see S1 Text). Bayesian phylogenetic models revealed significant positive relationships between our scores and all 3 metrics (Fig 2 and Table B in S2 Text). The amount of variation explained was relatively low for testes mass ($R^2 = 0.30$), stronger for Bateman gradients ($R^2 = 0.53$), and stronger still for $I_S$ ($R^2 = 0.97$), which provides a more accurate index of pre-copulatory sexual selection (see Discussion).

## Geographical gradients of sexual selection

To visualise global patterns, we mapped sexual selection scores for 9,836 bird species with geographical range data using standard GIS approaches (see Methods). This procedure generates mean sexual selection scores for all species occurring in each grid cell, revealing lower average levels of sexual selection in tropical regions (Fig 3A). To account for spatial autocorrelation among neighbouring grid cells, we modelled the relationship between latitude and average sexual selection for each cell using spatial simultaneous autoregressive (SAR) models. The results confirmed that sexual selection follows a latitudinal gradient, increasing significantly towards higher latitudes (Fig 3B). However, when we used the same procedure to assess geographic biases in knowledge, we found that average data certainty also increased in the temperate zone, with sexual selection scores being particularly robust in North America and Europe, where many bird species have been studied intensively for decades (Fig 3C and 3D).

These patterns reflect a clear correlation between improved data quality and higher sexual selection scores (S2 Fig), suggesting that geographical variation in ornithological knowledge could potentially explain the overall latitudinal gradient in sexual selection. This does not appear to be the case, however, because the gradient in sexual selection remains strongly significant when we restrict the analysis to 7,592 species with moderate to high data certainty

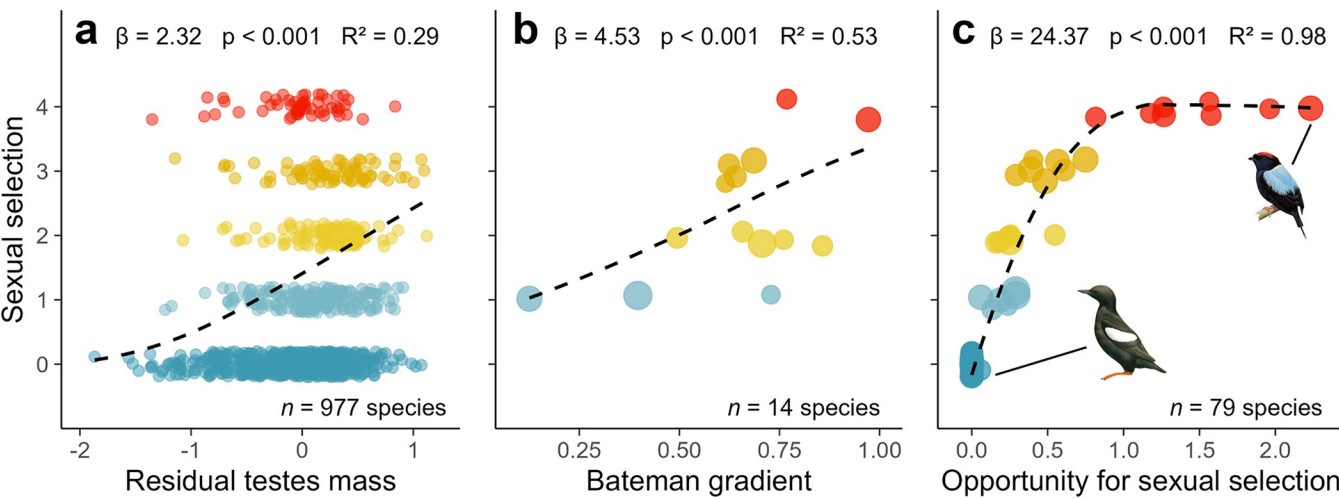

**Fig 2. Comparison of scores with alternative metrics of sexual selection.** Panels show results of Bayesian phylogenetic models assessing the relationship between sexual selection scores compiled for this study and 3 independent measures of sexual selection: (a) residual testes mass, (b) Bateman gradients ($\beta_{ss}$), (c) the opportunity for sexual selection ($I_s$) (see Methods). Dashed lines show the relationship for subsets of species with available data, using predictions from models accounting for phylogeny using a sample of 50 phylogenetic trees extracted from www.birdtree.org [50], grafted to the Prum and colleagues [61] genomic backbone. Effect sizes, $p$-values, and explained variance ($R^2$) were calculated from fixed effects only, and therefore represent the specific variation explained by each predictor separate from phylogenetic effects. Data-point colour reflects increasing levels of sexual selection from low (blue) to high (red). In (b) and (c), point size is scaled by the number of individuals used to calculate sex-specific metrics. To compare sex-specific metrics with bidirectional scores, we selected the largest $\beta_{ss}$ or $I_s$ from each species, treating males and females metrics equally. We estimated $I_s$ of 51 genetically monogamous species (with 0% extra-pair paternity) as zero. Excluding these species from $I_s$ models produced similar results (Table B in S2 Text). Illustrations show exemplars of extremely low (Black Guillemot) and extremely high (Lance-tailed Manakin) $I_s$, respectively; images are from Birds of the World, reproduced with permission of Cornell Lab of Ornithology. The data underlying this figure can be found at https://doi.org/10.6084/m9.figshare.27255609.

(scored 3–4; S3A and S3B Fig). Even when we restricted the analysis to the highest category of certainty ($n = 2{,}851$ species), the positive relationship between sexual selection and latitude remained, albeit weaker because highly polygamous tropical species such as cotingas and manakins are more easily classified with certainty than monogamous species, explaining the accumulation of high-certainty species in the tropics (S4 Fig) and the relatively high average sexual selection score among the best-known Amazonian birds (S3C and S3D Fig). The positive latitudinal gradient in sexual selection scores identified by our analyses may seem counter-intuitive given the occurrence of largely monogamous clades such as seabirds and raptors at higher latitudes, and reflects intense sexual selection reported in other temperate and polar species, including both passerines [62] and non-passerines [63–65].

### Univariate models predicting sexual selection

To understand the mechanisms driving latitudinal gradients in sexual selection, we re-examined geographical patterns through the lens of ecological traits. This approach revealed contrasting gradients when comparing primary consumers (herbivores, frugivores, granivores, and nectarivores) with secondary consumers (omnivores, carnivores, and scavengers). The positive latitudinal gradient in sexual selection is removed (nonsignificant) in primary consumers (Fig 4A and 4B) and strongly reversed (negative) in an ecologically important subset: frugivores (Fig 4C and 4D). Conversely, we found a strong positive gradient in secondary consumers (Fig 4E and 4F), particularly the invertivores, which comprise approximately 40% of the world's birds (Fig 4G and 4H). Thus, the overall positive gradient across all birds is predominantly driven by the species-rich invertivores, concealing an opposite (negative) gradient

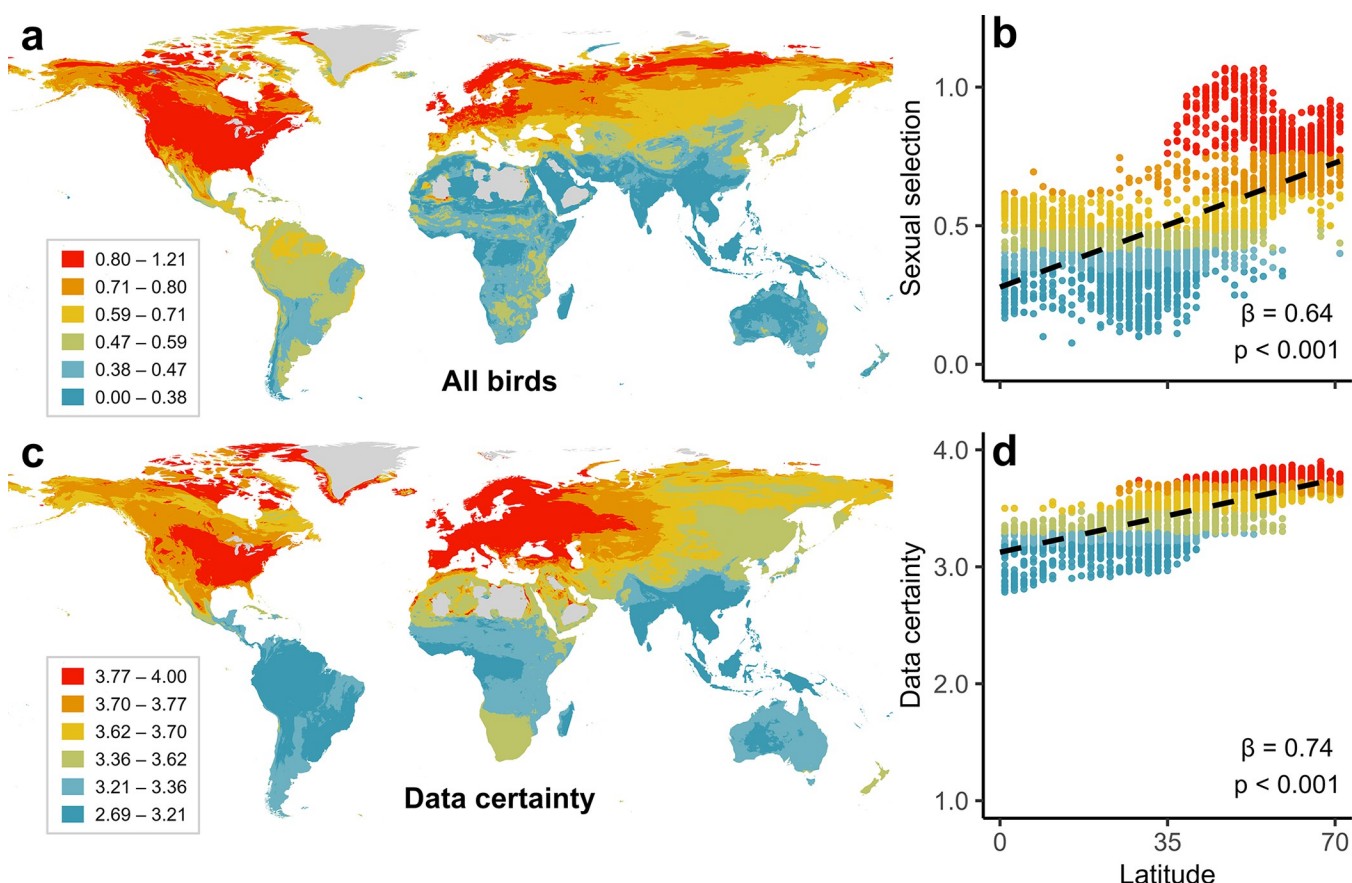

**Fig 3. Geographical distribution of sexual selection in birds.** (a) Worldwide variation in sexual selection scores for 9,836 bird species included in a global phylogeny (www.birdtree.org [50]), averaged for all species occurring in 5-km grid cells based on breeding range maps. Sexual selection was scored based on mating behaviour from monogamy (scored 0) to extreme polygamy (scored 4; Table 1). (b) Relationship between avian sexual selection and breeding latitude. (c) Geographical variation in data certainty, ranging from no evidence (scored 1) to strong evidence (scored 4; see Table A in S2 Text). (d) Relationship between data certainty and latitude. In (b) and (d), points denote mean sexual selection per 200-km grid cell (Behrmann projection); dashed lines were generated from spatial simultaneous autoregression (SAR) models predicting mean sexual selection strength. Additional SAR models on a subset of species with higher-quality data (scored 3–4 for data certainty) showed similar patterns (S1 Text and Table C in S2 Text). To reduce noise, cells with <10 species were excluded from all plots and models. To aid visualisation, plots were coloured using discrete intervals with an equal number of cells. Results are plotted using geographical range polygons provided by BirdLife International (www.datazone.birdlife.org) cropped to Earth's land-surface using the BIO1 climate layer (www.chelsa-climate.org). The data underlying this figure can be found at https://doi.org/10.6084/m9.figshare.27255609.

in fruit-eating species (frugivores), which constitute a smaller proportion (approximately 10%) of global bird diversity.

Switching focus to another key ecological trait, migration, we found that both migratory and non-migratory species showed a significant positive gradient in sexual selection across latitude (S5 Fig), aligning with the general trend observed in all birds. A similar positive trend was also found in territorial species, with sexual selection again increasing towards high latitudes from a low baseline in the tropics (S6 Fig). The gradient for non-territorial species was different, either flat when we accounted for spatial autocorrelation in SAR models (S6 Fig) or reversed when we modelled sexual selection at the species level using Bayesian regression models (S7 Fig). In all other cases, the results of our SAR models were replicated by species-level models (S7 Fig) and remained unchanged when we restricted analyses to well-known species (Table C in S2 Text), indicating that our results are not biased by modelling approach or data certainty. These highly consistent results suggest that a combination of ecological strategies relating to diet, migration, and resource defence shape global trends in avian sexual selection.

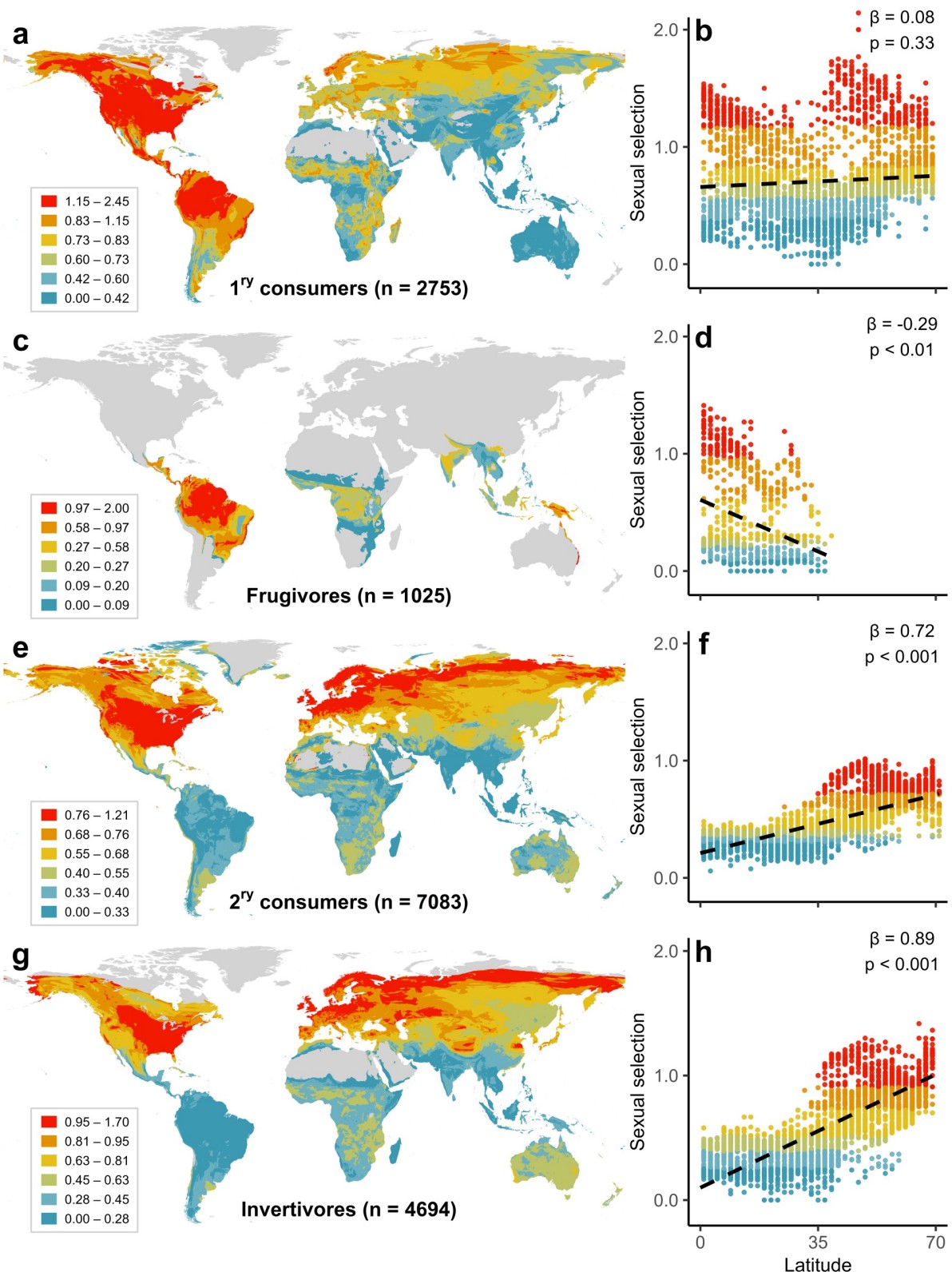

**Fig 4. Global distribution of sexual selection partitioned by trophic level and diet.** Upper panels show strength of sexual selection in primary (1^ry) consumers mapped globally (a) and plotted against latitude (b), as well as for frugivores only (c, d). Lower panels show strength of

sexual selection in secondary (2$^{ry}$) consumers mapped globally (e) and plotted against latitude (f), as well as for invertivores only (g, h). We treated omnivores, carnivores and scavengers as secondary consumers. Sexual selection was scored from monogamy (0) to extreme polygamy (4; see Methods). For mapping, we averaged sexual selection scores in each 5-km grid cell based on all species with breeding ranges overlapping the cell. To aid visualisation, maps were coloured using discrete intervals with an equal number of cells. In scatterplots, points represent mean sexual selection per 200-km grid cell; dashed lines were generated from spatial simultaneous autoregression (SAR) models predicting mean sexual selection strength. Sensitivity analyses on a subset of species with higher-quality data (scored 3–4 for data certainty) showed similar patterns (Table D in S2 Text). To reduce noise, cells with <10 were excluded from all plots and models. Results are plotted using geographical range polygons provided by BirdLife International (www.datazone.birdlife.org) cropped to Earth's land-surface using the BIO1 climate layer (www.chelsa-climate.org). The data underlying this figure can be found at https://doi.org/10.6084/m9.figshare.27255609.

However, the relative importance of each trait is difficult to determine based on raw patterns given our finding that sexual selection is phylogenetically conserved in birds (Fig 5A).

To evaluate the relative role of species traits while accounting for phylogenetic non-independence, we re-examined the correlation between traits and sexual selection using Bayesian phylogenetic models. As a first step, we ran a species-level model including trophic level as a

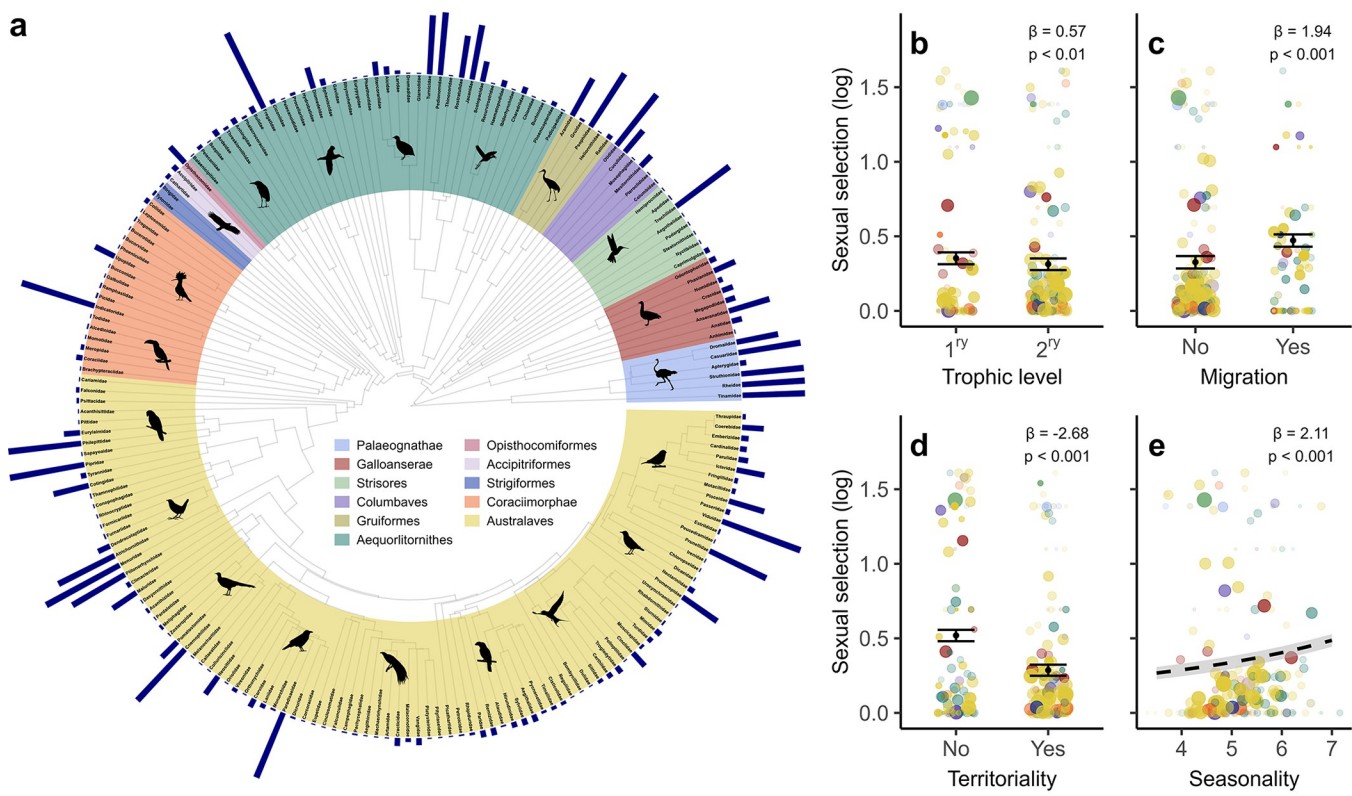

**Fig 5. The role of climate in regulating avian sexual selection.** (a) A family-level consensus phylogenetic tree showing the distribution of sexual selection scores across 194 families (*n* = 9,988 species). Terminal bars show average strength of sexual selection estimated for each family; longer bars indicate higher scores (see Methods). Coloured segments demarcate major clades; icons depict representative families. Panels (b–d) show average sexual selection scores for each family, partitioned by trophic level (b), migratory behaviour (c), and territoriality (d); black points show predicted mean values for each group from species-level Bayesian phylogenetic models; whiskers show 95% credible intervals. In (b), 1$^{ry}$ = primary; 2$^{ry}$ = secondary. Species traits shown in (b–d) are all hypothetically linked to temperature seasonality, which is also related to sexual selection scores (e); dashed line shows model fit from a species-level Bayesian phylogenetic model; shaded area denotes 95% credible intervals from model predictions. In (b–e), data points for each family are coloured by major clades (see phylogeny), with size and transparency of each data point scaled to within-family species richness. Statistics show effect size and *p*-value comparing main effects with reference groups: secondary consumer, no migration, and no territoriality, respectively. For ease of visualisation, data shown in panels are family-level averages, which do not fully reflect the stronger species-level correlations reported in statistics. Running the same models on high-certainty data produced similar results (Table F in S2 Text). The data underlying this figure can be found at https://doi.org/10.6084/m9.figshare.27255609. Icons were generated using R (rphylopic package) and are under the Creative Commons Attribution 4.0 International (CC BY 4.0) License.

binary predictor, with phylogenetic co-variance as a random effect (see Methods). We then repeated the same univariate model for migration and territoriality. All 3 ecological traits emerged as significant predictors (Tables E and F in S2 Text), with the intensity of sexual selection increasing in primary consumers (Fig 5B), migratory species (Fig 5C), and non-territorial species (Fig 5D). These ecological traits are all strongly tied to climatic seasonality [20,66], which we also found to be a powerful predictor of global trends in sexual selection score (Fig 5E and Tables E and F in S2 Text). This raises 2 important questions: first, whether any particular ecological trait is driving the overall patterns, and second, whether climatic factors play an important role, either directly or via their effect on species ecology.

## Multivariate models

To examine the relative roles of ecological and climatic factors, we included all predictors in a multivariate Bayesian phylogenetic model (Fig 6 and Table E in S2 Text) along with 2 key interactions accounting for complex relationships between diet and territoriality [20] and between diet and seasonality [67]. We found that the strongest constraint (negative effect) on sexual selection was territoriality, with significantly lower levels of sexual selection in species defending seasonal or year-round territories. Conversely, the strongest driver (positive effect)

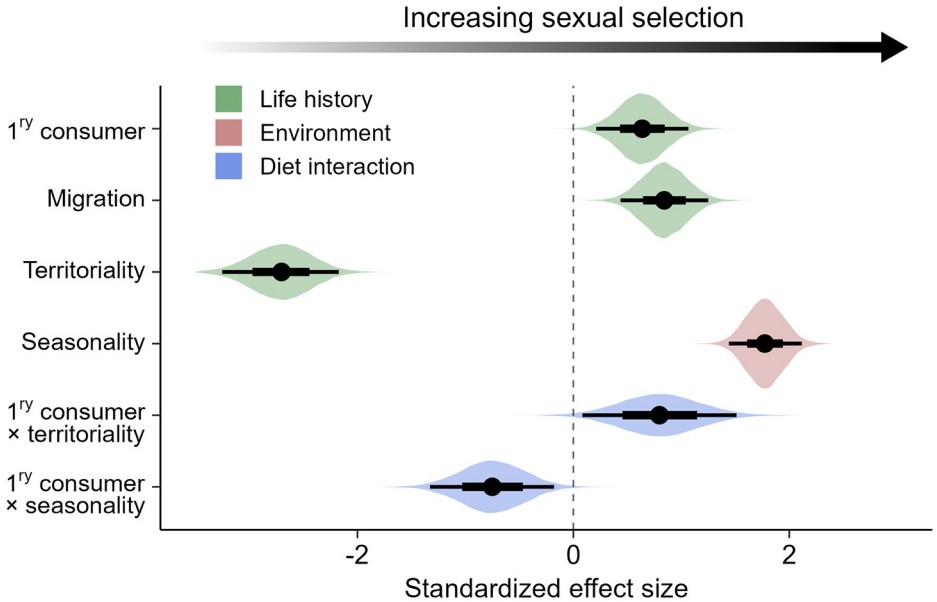

**Fig 6. Relative roles of ecology and climate as drivers of sexual selection in birds.** Results shown are from Bayesian phylogenetic models testing drivers of sexual selection in 9,836 species. Predictors include 3 life history variables (green), 1 climatic variable (pink), and 2 key interactions between diet and the dominant effects (territoriality and seasonality; blue). 1ry consumer = primary consumer. The reference groups for the 3 categorical predictors are as follows: secondary consumer; no migration; and no territoriality, respectively (see Methods for definitions). Models were run on a sample of 50 phylogenetic trees extracted from www.birdtree.org [50], grafted to the Prum and colleagues [61] genomic backbone. Dots show mean effect size estimates from 12,500 posterior draws. For each effect, broad bases of whiskers show 66% credible intervals; narrow tips of whiskers show 95% credible intervals. Coloured distributions indicate the spread of effect size estimates, generated from a sample of 1,000 posterior draws. Full statistical results are presented in Table F in S2 Text. The data underlying this figure can be found at https://doi.org/10.6084/m9.figshare.27255609.

was climatic seasonality, with significantly higher levels of sexual selection in species breeding in the most variable climates. While accounting for both these effects, we found that trophic level and migration also retained their independent significant positive effects on avian sexual selection (Fig 6 and Table E in S2 Text).

Although the independent effect of trophic level on sexual selection was relatively small, this ecological trait appears to have a key role in shaping variation in sexual selection among species through significant interactions with both territoriality (β = 0.80, 95% CI = 0.08, 1.51) and climatic seasonality (β = −0.75, 95% CI = −1.33, −0.18; Fig 6 and Table E in S2 Text). In other words, dietary factors regulate the effects of territoriality and seasonality, which in turn have different implications for species at different trophic levels. We found that a phylogenetic model incorporating all ecological traits, along with the effects of seasonality and the key interactions with diet, explained a very high proportion of variation in sexual selection ($R^2$ = 0.90; 95% CI = 0.88, 0.92). The proportion of variance explained was equally strong ($R^2$ = 0.92; 95% CI = 0.91, 0.94) when we ran a more conservative analysis focusing exclusively on species with high-certainty data (S8 Fig and Table F in S2 Text).

## Discussion

Based on a newly compiled global data set, our analyses confirm that the strength of sexual selection is subject to opposing latitudinal patterns in species with different ecologies. Specifically, the intensity of selection increases towards higher latitudes in secondary consumers, with contrastingly flat or reversed gradients in primary consumers (Fig 4). These findings explain why positive latitudinal gradients of sexual selection tend to be proposed by authors studying insectivorous birds [16–18], whereas negative gradients are proposed in studies of frugivores [22,23]. Differences across ecological groups may also help to resolve conflicting reports in the literature from a wider range of animal systems, with putative latitudinal gradients in sexual selection varying from positive [31,56] to negative [15,68], as well as many studies reporting a lack of significant gradients when averaging across all species [27–30].

To unravel the mechanistic drivers of these complex and inconsistent global trends, we ran phylogenetic models showing that variation in sexual selection across birds is predominantly explained by 2 interlinked factors—temperature seasonality and territoriality—both of which vary strongly with latitude. The strongest positive predictor of sexual selection across all birds was seasonality, in line with previous studies identifying a link between sexual selection and climatic heterogeneity, including variation in temperature [33], rainfall [56], or even local weather patterns [9]. An important consequence of climatic heterogeneity, including seasonality, is that key phenomena such as plant productivity and associated food tend to cluster in time [69]. As a result, breeding individuals experience more intense competition for available mates during a shorter reproductive window [70], as well as a concentrated availability of mates and more efficient multiple matings [19]. However, our findings clarify that climatic seasonality does not act alone, but in concert with other factors that can regulate or even reverse geographic gradients in sexual selection.

Among the ecological traits assessed in our analyses, territoriality was the strongest predictor of sexual selection, with a strong negative effect. This correlation may reflect a tendency for territoriality and monogamy to coevolve [71] or perhaps arises because the defence of non-overlapping territories automatically constrains population density and therefore limits access to additional mates and opportunistic matings. Davies and Lundberg [55] illustrated this point by manipulating resource density in the territories of female Dunnocks (*Prunella modularis*), showing reduced territory size and higher rates of polygamy as access to resources increased. Territories can have different implications for nectarivores, in which males of some species defend

patches of flowers to attract potential mates [72]. Similarly, in mammalian systems, variation in social dominance can lead to variation in territory quality among males [73], and consequently to increased mating success for those with the best territories (known as resource defence polygyny). While this may cause a positive link between territoriality and sexual selection in some avian systems, we find that the overall global pattern is reversed, presumably because resource defence polygyny is much rarer than stable monogamy, at least in tropical birds [20].

To examine how the effects of climatic seasonality and resource defence strategies are influenced by diet more generally, we included interactions with trophic niches in our phylogenetic models. We found that trophic niche had significant interactions with both seasonality and territoriality, indicating that diet and foraging ecology plays a crucial role in shaping geographic variation in sexual selection. On the one hand, climatically stable environments such as tropical rainforests may be associated with increased sexual selection in primary consumers because the year-round abundance of nutrient-rich fruit and flowers favours breeding systems in which males are freed from the constraints of territory defence and parental care (leading to extreme polygyny in lekking frugivores and nectarivores). Year-round territoriality increases slightly in tropical frugivores but remains rare overall (S9 Fig). On the other hand, in secondary consumers such as insectivores, climatic stability promotes stable year-round pair- or group-territoriality (S9 Fig), reducing the opportunity for sexual selection in rainforest regions. These stable territorial systems are largely absent at temperate latitudes or in polar regions, where most insectivores are subject to much higher levels of sexual selection on ephemeral breeding territories before migrating to lower latitudes during winter [70,74].

We included migration in our models because migratory behaviour correlates with sexually selected traits and breeding systems across a range of taxa [32,75–77]. Even within single species, female choice has been shown to exert stronger sexual selection in migratory than sedentary populations [78]. Despite the widespread consensus that migration promotes sexual selection, the effects can mostly be ascribed to climatic seasonality, which is very strongly correlated with migratory behaviour in birds [79]. Indeed, seasonal climatic fluctuations are the fundamental reason for the near-absence of insect prey, flowers, and fruit during the winter period at higher latitudes, forcing many species to migrate [74]. Intriguingly, our models reveal that, even after accounting for the effect of seasonality, migration remains weakly but significantly associated with sexual selection. This independent effect of migration is difficult to explain, perhaps relating to higher population density or more extreme breeding synchrony than resident species living in highly seasonal environments [80].

## Sexual selection in the context of climate change

Our findings indicate that geographical variation in sexual selection may be sensitive to shifts in seasonality projected to occur worldwide and particularly in mid- to high latitudes [81]. In mid-latitudes, reduced climatic fluctuations and milder winters may lead to increased survival of resident birds, in parallel with year-round availability of insects, nectar, and fruit. This can reduce selection for migration [74], causing migratory bird populations to become sedentary [82], while also increasing the duration of breeding seasons for resident species [83]. Our results suggest that these behavioural changes will dampen sexual selection, in line with previous studies showing that higher temperatures [77] and aseasonal climates [84,85] can modify competition for mates and mating opportunities. Nonetheless, the precise impacts of climate change on mechanisms of sexual selection and their adaptive implications [4,86–88] are hard to predict because lengthening warm seasons have variable and species-specific effects on reproductive phenology in both plants [89] and animals [83,90].

### Sexual selection scores for all birds: Validity, robustness, and research priorities

Our analyses offer the most comprehensive synthesis of geographical variation in sexual selection, estimated for 10,671 bird species, including all 9,988 valid species aligned with a global phylogeny [50]. We provide this information on sexual selection, data sources, and data certainty for all species in S1 Data, representing a major step change in data availability from the most recent compilations of sexual selection scores, covering 3,250 passerine species [32,48]. We extended these earlier data sets, not only in terms of species coverage, but also by scoring sexual selection based on a wider accumulation of evidence, including molecular evidence of polygamy and extra-pair paternity.

We found that these revised scores correlated relatively weakly with a widely used metric of sexual selection in birds: residual testes mass (Fig 2A). One reason for this weak relationship is that residual testes mass appears to decline in species with the highest levels of sexual selection. This makes sense because females tend not to remate in lekking systems [91] and males consequently invest heavily in pre-copulatory sexual competition [92]. Thus, testes mass reflects sperm competition but is a relatively crude metric for sexual selection. In contrast, our scores correlated more strongly with 2 other metrics—Bateman gradient and $I_S$ (Fig 2B and 2C)—which provide a more complete estimate of the strength of sexual selection. Overall, these correlations suggest that our scores provide a robust metric of variance in mating success, at least in cases where sufficient evidence is available.

Our results also reveal a strong latitudinal gradient in data certainty (Fig 3C and 3D), reflecting a long-standing bias towards ornithological research in the temperate zone [70] as well as the difficulty of tracking down relevant literature published in languages other than English. However, we note that most high-certainty species in our data set are found in the tropics (S4 Fig), suggesting that biases in knowledge to some extent reflect the sheer number of tropical species compared with higher latitudes. The fact that high-certainty scores are distributed across all latitudes (S4 Fig) is crucial, allowing us to re-run our analyses with a more conservative data set, confirming that all our main results are robust to biases in knowledge (Tables C–F in S2 Text).

We hope that the open release of comprehensive estimates of sexual selection for over 10,000 bird species provides a template for further macro-scale research. General hypotheses linking sexual selection to evolutionary mechanisms and ecological contexts [93,94] can now be examined at unprecedented scale. We also hope our global data set can re-energise efforts to investigate avian mating systems in data-poor regions. In particular, the distribution of bird species with low-certainty data (Figs 3C and S4) offers a detailed map of research priorities, highlighting the urgent need for observational and molecular studies, especially in the tropics, to quantify polygamy and EPP in species with low to moderate levels of sexual selection.

### Conclusions

Our analyses show that global-scale gradients in avian sexual selection are driven largely by temperature seasonality, with the strength of selection also strongly constrained by territoriality, promoted by migration, and modified by interactions with dietary niche. This complex but consistent interplay of environmental factors and species traits helps to clarify some of the fundamental yet elusive eco-evolutionary processes driving global variation in mating systems and sexual traits in animals [12]. On one hand, we identify context-dependent associations that provide a clear explanation for the conflicting latitudinal patterns reported in previous studies, thus helping to resolve a long-standing debate about the validity of latitudinal gradients in sexual selection and their underlying mechanisms [13,15]. More broadly, these analyses establish a

standardised metric for addressing major research objectives, including the role of sexual selection in speciation [3] and the evolutionary impacts of environmental change [82,95,96].

## Methods

### Quantifying sexual selection

We scored sexual selection for all bird species using information published in primary and secondary literature, building on previous estimates of sexual selection [32,48,58,97]. Most information at family- and species-level was extracted from regional or global handbooks, including the Handbook of the Birds of the World series [98] with recent updates [99]. When published information was inconclusive, we used expert knowledge to guide scoring decisions.

To assign scores, we adapted a well-established system based on the estimated proportion of polygamous individuals of either sex in the population [32,39–41]. By our definition, monogamy involves both sexes maintaining the same breeding partner for the duration of a reproductive event or season, and thus includes both sequential and perennial (lifelong) monogamy. In data-poor cases, we used observations of breeding behaviour, including pair-territoriality and offspring provisioning roles, to infer the degree of social monogamy and polygamy. In addition, we further developed previous approaches by integrating information on EPP and display behaviour where possible. We assumed that high rates of EPP are associated with larger variance in male reproductive success [21], and thus reflect elevated levels of sexual selection even when species are socially monogamous. Assignments were made based on quantitative estimates or textual descriptions of observed polygamy, extra-pair behaviour, or behavioural displays, with the addition of genetic estimates of EPP data, now available for a growing number of species [49].

Using the full set of criteria described in Table 1, we assigned sexual selection scores of 0 (strict monogamy), 1 (frequent monogamy), 2 (regular polygamy), 3 (frequent polygamy), or 4 (extreme polygamy) to all species. We assumed that sexual selection is lowest in strict monogamy (0) and highest for extreme polygamy (4). Allocation to the first 4 categories was based on thresholds for polygamy suggested by previous studies: <0.1%, 0.1%–5%, 5%–20%, and >20% for scores of 0–3, respectively [32,39–41]. We used similar but higher quantitative thresholds for EPP data for 2 reasons. First, species may be strictly monogamous yet still have reported EPP rates above zero because of factors such as mate switching or brood parasitism [100]. Consequently, following previous authors [101], we define monogamy as <5% EPP. Second, many socially monogamous species have relatively high levels of EPP, so we adjusted thresholds upwards (assigning populations with 5%–25%, 25%–50%, and >50% EPP to sexual selection scores 1–3, respectively). This avoids inflating sexual selection scores for species with EPP data, a biased subset of well-studied birds mainly inhabiting the temperate zone [49]. Finally, species with highly elaborate mating display behaviours, including social lekking (e.g., Birds-of-Paradise, Paradisaeidae) or permanent display locations (e.g., bellbirds, *Procnias*), were given the highest score (4) because variance in the reproductive success of males is likely to reach its peak in these systems [102–104]. Note that our scoring system does not assume correlation between different components of our score because, for example, polygamous species may have low EPP while species with high levels of EPP may be socially monogamous. Instead, we assigned the highest sexual selection scores triggered by any criterion.

Unlike previous bidirectional scoring systems [32], we treated sexual selection as a unidirectional variable. That is, we make no distinction between male or female-biased mating systems, instead grouping polyandry and polygyny under the general definition of polygamy. To facilitate any future research that relies on discriminating between female-biased and male-biased sexual selection, we provide an updated list of sex-role-reversed bird species (S1 Data)

so that a bidirectional score can be generated by reversing the sign on scores for sex-role-reversed species.

To enable phylogenetic analyses, we aligned sexual selection scores and associated geographical range data to the most comprehensive phylogeny available [50]. The taxonomy used in this phylogeny requires updating, so we also provide an additional data set of sexual selection scores aligned with 10,671 species included for a more recent taxonomy [57]. Although we do not use this data set in our own analyses, we hope that it will enable future studies using forthcoming phylogenetic data sets [105], as well as several rich sources of information, including eBird citizen-science data [106], Birds of the World [99], and AVONET [52], all of which use the same updated taxonomic format [57]. Sexual selection scores under both taxonomic treatments, along with the sources of information supporting these assignments, are presented in S1 Data.

### Testing sexual selection metrics

To validate the accuracy of our proposed scoring system in estimating sexual selection, we conducted a comparative analysis against 3 widely used measures of sexual selection from existing published data sets: residual testes mass [97,107], Bateman gradients ($\beta_{SS}$) [108], and the opportunity for sexual selection ($I_S$) [45] (for a full explanation, sources and derivation of these metrics see S1 Text). Given the scarcity of studies estimating sexual selection, we also derived additional estimates of zero $I_S$ from studies that recorded 0% EPP rates, as summarised by Brouwer and Griffith [49]. We assessed the strength of association between our scores and each metric using Bayesian phylogenetic models (S1 Text), with each metric as a single predictor. To test the accuracy of each metric, we extracted standard $p$-values for the strength of association, and calculated marginal $R^2$ values to determine how closely scores and metrics aligned. To confirm a significant relationship with $I_s$ was due to estimates from species with 0% EPP, we performed a sensitivity analysis excluding the 51 species with estimated $I_s$ scores, which produced results consistent with the full models, as presented in Table B in S2 Text.

### Accounting for biases in data certainty

Assembling data on sexual selection is challenging because published information for many species is sparse or based on few observations. Basic details of breeding behaviour are now available for almost all birds (e.g., [99]), but the format, quality, and relevance of information varies greatly across species. To address these biases, we scored data certainty using a system adapted from Tobias and colleagues [20], from 1 (lowest certainty) to 4 (highest certainty; see Table A in S2 Text for full definitions). Species lacking direct observations receive higher certainty scores if they belong to taxonomic groups with consistent and well-documented behaviour (e.g., a poorly known species of sandgrouse would be scored 3 because all known Pteroclidae are monogamous with biparental care). Species in groups with some variation in breeding systems would instead be scored 2 because the degree of confidence in inferred data is reduced.

### Ecological variables

To assess the role of evolutionary drivers, we extracted data on species-specific traits from published data sets [20,52]. We selected 3 fundamental ecological traits previously proposed to influence levels of sexual selection: trophic level (linked to both spacing and constraints on parental care [22,53,109]), migration (linked to breeding synchrony or constraints on the duration of breeding seasons [54]), and territoriality (linked to movement and spacing patterns [55]).

To facilitate the comparison of categorical and continuous data, we converted trophic level, migration, and territoriality to binary variables [110]. For trophic level, we group species as either obligate primary consumers, defined as those with more than 60% of their energetic requirements provided by plant material, or secondary consumers, including omnivores, carnivores, and scavengers from higher trophic levels. This was because primary consumers generally experience a more heterogeneous landscape than higher trophic levels, often with scarce resources of high nutritional value. Furthermore, previous research has shown a potential relationship between herbivory and polygamy in mammals [111] that may also be present within birds.

Migration was scored in 3 categories: sedentary, partially migratory (minority of population migrates long distance or most individuals migrate short distances), and migratory (majority of population undertakes long-distance migration). We dichotomized migration by grouping sedentary and partial migrants together, because obligate long-distance migrants tend to maintain consistent departure times, distances, and directions [112], which should all lead to higher breeding synchronicity. Territorial behaviour was classified as either: "none" (never territorial or at most defending very small areas around nest sites), "weak" (weak or seasonal territoriality, including species with broadly overlapping home ranges or habitually joining mixed species flocks), and "strong" (territories maintained throughout year). We dichotomized territoriality by grouping weak and strong territorial behaviour together, as both potentially create strong spatial constraints that in turn may influence the mating opportunities [55,113].

## Climatic seasonality

To account for the effects of climate seasonality in models of sexual selection, we included species-specific data on average temperature seasonality extracted from the CHELSA bioclim data set v2.1 [114]. We selected temperature seasonality (bio4) as our primary measure of climate variability because temperature has been shown to predict mating systems and sexual selection [77,115]. The key metric (bio4) provides an index of local intra-annual temperature variation, with grid cells equal to the standard deviation of monthly mean temperatures across each year, using global data from 1981 to 2010 at 30 arcsecond resolution. To extract an average seasonality estimate for each species, we overlaid climate data with expert-drawn breeding ranges provided by BirdLife International [116] (S1 Text). Grid cells that fell within each species' breeding range were averaged using a Behrmann equal area projection, disregarding cells with less than 50% overlap. Removing species without range maps, this left 9,836 species for subsequent statistical analysis.

## Statistical analysis

To identify latitudinal gradients in sexual selection, we first quantified mean sexual selection score for each grid cell using a 200 km resolution Berhmann equal area projection (S1 Text). Grid cell values were calculated from all bird species with available geographical range data ($n = 9,836$). We then modelled mean sexual selection using spatial SAR models from the R package *spatialreg* [117], with absolute latitude (distance from the equator) as the sole predictor. Next, to identify any latitudinal bias in data certainty, we ran similar models on the same sample of species with their mean data certainty score as the response variable. Given a strong correlation between latitude and data certainty (Fig 3C and 3D), we ran subsequent latitudinal models predicting sexual selection for higher quality data only. In this case, we limited the sample to species with moderate to high data certainty (scored 3–4; $n = 7,592$ species) and also ran a maximally conservative model restricted to high-certainty data only (scored 4; $n = 2,851$ species).

To determine how species ecology shapes global trends, we ran subsequent latitudinal models under the following partitions: primary consumers ($n = 2,753$), frugivores ($n = 1,025$),

secondary consumers ($n$ = 7,083), invertivores ($n$ = 4,694), long-distance migrants ($n$ = 901), resident and short-distance migrants ($n$ = 8,935), territorial species ($n$ = 7,261), and non-territorial species ($n$ = 2,575). To account for spatial differences in species richness, we also ran Bayesian regression models (species-level) from the R package *brms* [118], using the centroid latitude of each species' range as the predictor (S1 Text). Given the ordinal nature of our sexual selection scores, we used a cumulative family distribution, which can be interpreted similarly to standard generalised linear models [119]. To account for variability in data quality, we also repeated each set of models on the subset of species with moderate to high data certainty (certainty scores 3 and 4). It was not possible to restrict these models to species scored 4 for data certainty because of low sample size in many categories.

After identifying raw latitudinal gradients in sexual selection, we used univariate and multivariate Bayesian phylogenetic models to link potential evolutionary drivers with the strength of sexual selection across species. Preliminary analyses show that the strength of sexual selection is highly conserved in birds (see S1 Text). Consequently, we included a phylogenetic covariance matrix as a random effect, using the Jetz and colleagues [50] tree topology grafted to the Prum and colleagues [61] genomic backbone. To ensure our results are robust to uncertainty in tree topology, we repeated each model over 50 randomly selected trees, combining draws into a single posterior distribution.

To determine the role of species ecology in directly predicting sexual selection, we ran phylogenetic models incorporating information on trophic level, migration, territoriality, and seasonality. As the only continuous variable, seasonality was log-transformed to approximate normality and standardised to 2 standard deviations to facilitate comparison with categorical predictors [110]. All categorical predictors were centred on zero to reduce the collinearity with interaction terms [120]. To address the hypothesis that spatial and temporal heterogeneity of resource abundance drive sexual selection, we also included trophic level interactions with seasonality (temporal heterogeneity) and territoriality (spatial heterogeneity). All predictors and interactions had variance inflation factor (VIF) values below 3 (Table G in S2 Text), indicating that collinearity would not affect model interpretation. To account for biases in data quality, we re-ran univariate and multivariate analyses on higher quality data (certainty 3 or 4; $n$ = 7,592). All analyses were conducted in R version 4.2.1 [121].

## Supporting information

**S1 Text. Supplementary information.** Supplementary methods, results and discussion. The data and code can be found at https://doi.org/10.6084/m9.figshare.27255609. (DOCX)

**S2 Text. Supplementary tables.** Table A in S2 Text. Scoring system for assigning data certainty to estimates of sexual selection in birds. **Table B in S2 Text.** Comparing alternative metrics of sexual selection. **Table C in S2 Text.** Latitudinal gradients in sexual selection. **Table D in S2 Text.** Species-level latitudinal gradients in sexual selection. **Table E in S2 Text.** Ecological predictors of sexual selection. **Table F in S2 Text.** Results of sensitivity analyses assessing robustness of models to data certainty. **Table G in S2 Text.** Collinearity between multivariate model predictors. (DOCX)

**S1 Fig. Species richness of birds assigned to different levels of sexual selection.** Plots show number of species in categories 0–4 according to the species limits in BirdTree [50] (a) or a more recent taxonomic update [57] (b), as well as within the 2 largest dietary guilds: invertivory (c) and frugivory (d). The strength of sexual selection increases from 0 (strict monogamy)

to 4 (extreme polygamy) according to our scoring system (Table 1). The higher species total in Clements is caused by the addition of a few newly described species along with several hundred taxonomic splits. The data underlying this figure can be found at https://doi.org/10.6084/m9.figshare.27255609.
(TIF)

**S2 Fig. Sexual selection strength across data certainty partitions.** Plots show how the certainty in data for each species affects (a) average sexual selection and (b) variation in sexual selection score. In (a) and (b), sexual selection is scored from 0 (strict monogamy) to 4 (extreme polygamy; see Table 1) and data certainty is scored from 1 (no direct or indirect evidence) to 4 (direct evidence published in primary and secondary literature; see Table A in S2 Text). In (a), points show average sexual selection for each category of data certainty; whiskers denote one standard error; sample sizes are the total number of species in each data certainty partition. In (b), stacked bars show the relative proportion of sexual selection scores across each category of data certainty. Bars are coloured according to sexual selection score (SS) (blue = 0; red = 4). The data underlying this figure can be found at https://doi.org/10.6084/m9.figshare.27255609.
(TIF)

**S3 Fig. Effects of data quality on the distribution of sexual selection.** Upper panels show average sexual selection for a subset of species with higher-quality data (scored 3–4 for data certainty; $n = 7,592$) mapped globally (a) and plotted against latitude (b). Lower panels show average sexual selection for a smaller sample of species ($n = 2,851$) in the top category for data certainty (scored 4), again mapped globally (c) and plotted against latitude (d). In a and c, averages for each cell are calculated from all species with breeding range maps overlapping each 5-km grid cell. To aid visualisation, maps were coloured using discrete intervals with an equal number of cells. In b and d, points represent mean sexual selection per 200-km grid cell; dashed lines were generated from spatial simultaneous autoregression (SAR) models predicting mean sexual selection strength. To reduce noise, cells with <10 species were excluded from all plots and models. Results are plotted using geographical range polygons provided by BirdLife International (www.datazone.birdlife.org) cropped to Earth's land-surface using the BIO1 climate layer (www.chelsa-climate.org). The data underlying this figure can be found at https://doi.org/10.6084/m9.figshare.27255609.
(TIF)

**S4 Fig. The geographical distribution of species with high-certainty data.** Worldwide species richness for 2,851 bird species with high data certainty (score = 4) included in a global phylogeny (www.birdtree.org [50]). Species were scored as 4 based on direct evidence published in primary and secondary literature (see Table A in S2 Text). Cell values represent the total number of high-certainty species, calculated by counting the number of breeding range maps overlapping more than 50% of each 5-km grid cell. To aid visualisation, cell values were grouped into equal-sized bins to reduce a skew in the colour scale caused by outlier cells with high species richness. Cells with <10 species were excluded. Results are plotted using geographical range polygons provided by BirdLife International (www.datazone.birdlife.org) cropped to Earth's land-surface using the BIO1 climate layer (www.chelsa-climate.org). The data underlying this figure can be found at https://doi.org/10.6084/m9.figshare.27255609.
(TIF)

**S5 Fig. Global distribution of sexual selection partitioned by migration tendency.** Based on occurrence data from the breeding range, upper panels show strength of sexual selection in long-distance migrants mapped globally (a) and plotted against latitude (b). Lower panels

show strength of sexual selection in short-distance and resident species mapped globally (c) and plotted against latitude (d). Sexual selection was scored in 5 categories ranging from monogamy (0) to extreme polygamy (4; see Methods). In maps (a, c), averages for each cell are calculated from all species with breeding ranges overlapping each 5-km grid cell. To aid visualisation, maps were coloured using discrete intervals with an equal number of cells. In scatterplots (b, d), points represent mean sexual selection per 200-km grid cell; dashed lines were generated from spatial simultaneous autoregression (SAR) models predicting mean sexual selection strength. Additional SAR models on a conservative data set (certainty scored 3–4) showed similar patterns (see Table C in S2 Text). To reduce noise, cells with <10 species were excluded from all plots and models. Results are plotted using geographical range polygons provided by BirdLife International (www.datazone.birdlife.org) cropped to Earth's land-surface using the BIO1 climate layer (www.chelsa-climate.org). The data underlying this figure can be found at https://doi.org/10.6084/m9.figshare.27255609.
(TIF)

**S6 Fig. Global distribution of sexual selection partitioned by territorial behaviour.** Upper panels show strength of sexual selection in seasonal and year-round territorial species mapped globally (a) and plotted against latitude (b). Lower panels show strength of sexual selection in non-territorial species mapped globally (c) and plotted against latitude (d). Sexual selection was scored in 5 categories ranging from monogamy (0) to extreme polygamy (4; see Methods). In maps (a, c), averages for each cell are calculated from all species with breeding range maps overlapping each 5-km grid cell. To aid visualisation, maps were coloured using discrete intervals with an equal number of cells. In scatterplots (b, d), points represent mean sexual selection per 200-km grid cell; dashed lines were generated from spatial simultaneous autoregression (SAR) models predicting mean sexual selection strength (see Methods). Additional SAR models on a conservative data set (certainty scored 3–4) showed similar patterns and are reported in Table C in S2 Text. To reduce noise, cells with <10 species were excluded from all plots and models. Results are plotted using geographical range polygons provided by BirdLife International (www.datazone.birdlife.org) cropped to Earth's land-surface using the BIO1 climate layer (www.chelsa-climate.org). The data underlying this figure can be found at https://doi.org/10.6084/m9.figshare.27255609.
(TIF)

**S7 Fig. Species-level latitudinal gradients in sexual selection.** Panels show results of Bayesian species-level regression models predicting variation in the strength of sexual selection across latitude, with species samples partitioned by ecological traits. To aid visualisation, species were pooled into 5-degree latitude bins, based on the centroid latitude of their breeding ranges. Points denote mean sexual selection, scaled by the relative sample size of each 5-degree bin; bars denote 95% credible intervals; dashed lines were generated from species-level regression models (see Methods). To reduce noise, latitudinal bins with <10 species were excluded from all plots; model predictions extend across all bins containing species data. Additional species-level models restricted to a subset of species with higher-quality data (scored 3–4 for data certainty) showed similar patterns (Table D in S2 Text). The data underlying this figure can be found at https://doi.org/10.6084/m9.figshare.27255609.
(TIF)

**S8 Fig. Evolutionary drivers of avian sexual selection.** Results shown are from Bayesian phylogenetic models testing drivers of sexual selection in species scored with high data certainty scores (scored 3–4; $n$ = 7,592 species). Predictors include 3 life history variables (green), 1 climatic variable (pink), and 2 key interactions between diet and the dominant effects

(territoriality and seasonality; blue). The reference groups for the 3 categorical predictors are as follows: secondary consumer; no migration; and no territoriality, respectively (see Methods for definitions). Models were run on a sample of 50 phylogenetic trees extracted from www.birdtree.org [50], grafted to the Prum and colleagues [61] genomic backbone. Dots show mean effect size estimates from 12,500 posterior draws. For each effect, broad bases of whiskers show 66% credible intervals (CI); narrow tips of whiskers show 95% CI. Coloured distributions indicate the spread of effect size estimates, generated from a sample of 1,000 posterior draws. Full statistical results are presented in Table G in S2 Text. The data underlying this figure can be found at https://doi.org/10.6084/m9.figshare.27255609.
(TIF)

**S9 Fig. Biogeography of avian territoriality.** Proportion of territorial primary consumers mapped globally (a), and plotted against latitude (b), compared with those holding year-round territories only (c, d). Proportion of territorial secondary consumers mapped globally (e) and plotted against latitude (f), compared with those holding year-round territories only (g, h). Territoriality data (see Methods) were converted into binary scores for mapping purposes: in a, b, e, and f (0 = none; 1 = seasonal/year-round); in c, d, g, and h (0 = none/seasonal; 1 = year-round). In maps (a, c, e, g), averages for each 5-km grid cell are calculated from all species with breeding range maps overlapping each cell. To aid visualisation, maps were coloured using discrete intervals with an equal number of cells. In scatterplots (b, d, f, h), points represent mean sexual selection per 200-km grid cell; dashed lines were generated from spatial simultaneous autoregression (SAR) models predicting mean sexual selection strength. To reduce noise, cells with <10 species were excluded from all plots and models. Results are plotted using geographical range polygons provided by BirdLife International (www.datazone.birdlife.org) cropped to Earth's land-surface using the BIO1 climate layer (www.chelsa-climate.org). The data underlying this figure can be found at https://doi.org/10.6084/m9.figshare.27255609.
(TIF)

**S1 Data. Species level sexual selection information for the world's birds. Sheet 1 in S1 Data. Metadata.** Definitions and sources for all variables in subsequent data sheets. **Sheet 2 in S1 Data. Sexual selection scores (BirdTree).** Sexual selection scores and associated uncertainty estimates and source citations for 9,988 bird species included in the BirdTree data set [50], along with corresponding taxonomic, ecological, morphological, and distributional data, where available (*n* = 9,836 species). **Sheet 3 in S1 Data. Sexual selection scores (eBird).** Sexual selection scores aligned with an updated taxonomy [57] used in major online resources, including eBird and Birds of the World (*n* = 10,671 species). **Sheet 4 in S1 Data. Sexual selection metrics.** Measured residual testes mass, Bateman gradients ($\beta_{SS}$), and Opportunity for Sexual Selection ($I_S$) values used in sensitivity analysis. **Sheet 5 in S1 Data. Taxonomic crosswalk.** Guide to species matching for users transferring data between BirdTree and eBird taxonomies, which often differ in spelling and treatment of allospecies. **Sheet 6 in S1 Data. Data sources.** Full references for sources cited in Sheet 2.
(XLSX)

**S2 Data. Data used in figures showing family-level or species-level averages. Sheet 1 in S2 Data.** Mean sexual selection scores for bird families partitioned by trophic level, as shown in Fig 5B. **Sheet 2 in S2 Data.** Mean sexual selection scores for bird families, partitioned by migratory behaviour, as shown in Fig 5C. **Sheet 3 in S2 Data.** Mean sexual selection scores for bird families, partitioned by territorial behaviour, as shown in Fig 5D. **Sheet 4 in S2 Data.** Mean sexual selection scores and temperature seasonality for avian families, as shown in Fig 5E. **Sheet 5 in S2 Data.** Extracted model predictions from species-level phylogenetic models predicting sexual

selection score according to univariate life history traits. Values correspond to model predictions overlaying family averages in Fig 5B–5E. **Sheet 6 in S2 Data**. Mean sexual selection scores for each data certainty partition, as shown in S2a Fig **Sheet 7 in S2 Data**. Mean sexual selection scores for different ecological groups at each five-degree latitudinal bin, as shown in S7 Fig. (XLSX)

## Acknowledgments

We thank Jane Amirthanayagam, Jessamine Badcock-Scruton, Celina Chien, Emily DuVal, Anita Kristiansen, Xiaoya Lian, Thomas Munro, and Trevor Price for their help in data collection and discussion of results.

## Author Contributions

**Conceptualization:** Robert A. Barber, Joseph A. Tobias.

**Data curation:** Robert A. Barber, Jingyi Yang, Chenyue Yang, Oonagh Barker, Tim Janicke, Joseph A. Tobias.

**Formal analysis:** Robert A. Barber.

**Investigation:** Robert A. Barber, Joseph A. Tobias.

**Methodology:** Robert A. Barber, Joseph A. Tobias.

**Resources:** Robert A. Barber, Joseph A. Tobias.

**Supervision:** Joseph A. Tobias.

**Validation:** Robert A. Barber.

**Visualization:** Robert A. Barber, Joseph A. Tobias.

**Writing – original draft:** Robert A. Barber, Joseph A. Tobias.

**Writing – review & editing:** Robert A. Barber, Jingyi Yang, Chenyue Yang, Oonagh Barker, Tim Janicke, Joseph A. Tobias.

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
