## [Editor Report · Decision Letter 0]

23 Jan 2024

Dear Rob, 

So sorry for the extreme slowness here - I really though this was going to be a more straightforward process, but we had some issues with one Academic Editor having a COI and another travelling; many thanks for your patience during this extraordinary delay. Thank you for submitting your manuscript entitled "Climate and ecology predict latitudinal trends in sexual selection inferred from avian mating systems" for consideration as a Research Article by PLOS Biology.

Your manuscript has now been evaluated by the PLOS Biology editorial staff, as well as by an academic editor with relevant expertise, and I'm writing to let you know that we would like to send your submission out for external peer review.

Once your full submission is complete, your paper will undergo a series of checks in preparation for peer review. After your manuscript has passed the checks it will be sent out for review. To provide the metadata for your submission, please Login to Editorial Manager (https://www.editorialmanager.com/pbiology) within two working days, i.e. by Jan 25 2024 11:59PM.

If you would like us to consider previous reviewer reports, please edit your cover letter to let us know and include the name of the journal where the work was previously considered and the manuscript ID it was given. In addition, please upload a response to the reviews as a 'Prior Peer Review' file type, which should include the reports in full and a point-by-point reply detailing how you have or plan to address the reviewers' concerns (I'm aware that you've already done this, but I've left the wording in regardless) 

Kind regards,

Roli

Roland Roberts, PhD

Senior Editor

PLOS Biology

rroberts@plos.org

---

## [Decision Letter · Decision Letter 1]

20 Mar 2024

Dear Rob,

Thank you for your patience while we considered your revised "portable peer review" manuscript "Climate and ecology predict latitudinal trends in sexual selection inferred from avian mating systems" for publication as a Research Article at PLOS Biology. Your revised study has been evaluated by the PLOS Biology editors, the Academic Editor and by two new reviewers. You'll see that while both reviewers were privy to the reviews from the previous journal, one of them opted to assess your manuscript afresh; both sets of comments seem helpful.

Reviewer #1 read the previous reviews, and recognises the basis for their criticisms, but tends to buy your arguments. He reveals his identity and says that he cannot judge taxon-specific issues. However, he makes several points that challenge your use of evidence from other taxa to discuss your avian findings (e.g. in terms of the interaction of diet and latitude). Reviewer #2 tells us that s/he deliberately did NOT read the reviews, so that they could make up their own mind. S/he says that it’s “an interesting and rigorous piece of work” but worries about the strength of support for your findings. The main issue is your reliance on very small p-values, despite the visually unconvincing “trends”; s/he wonders if you could make your case using visuals and effect sizes. S/he also wants you to be more cautious in your interpretation, flagging a potentially nonsensical or counter-intuitive result, and also asks if these patterns could arise through historical contingency rather than being driven by latitude.

In light of the reviews, which you will find at the end of this email, we would like to invite you to revise the work to thoroughly address the reviewers' reports.

Given the extent of revision needed, we cannot make a decision about publication until we have seen the revised manuscript and your response to the reviewers' comments. Your revised manuscript is likely to be sent for further evaluation by all or a subset of the reviewers.

**IMPORTANT - SUBMITTING YOUR REVISION**

*Re-submission Checklist*

*Published Peer Review*

*PLOS Data Policy*

Sincerely,

Roli

Roland Roberts, PhD

Senior Editor

PLOS Biology

rroberts@plos.org

REVIEWERS' COMMENTS:

Reviewer #1:

[identifies himself as Glauco Machado]

It was a great pleasure to read the manuscript entitled "Climate and ecology predict latitudinal trends in sexual selection inferred from avian mating systems". The text is extremely well-written, and the overall theoretical question is clearly presented in the introduction. I carefully went through the methods and was able to follow all the analytical procedures performed by the authors. Despite the complexity of the analyses, the results are presented clearly, and the figures are especially effective in illustrating the general patterns. The discussion, although focused on birds, is interesting and places the obtained results in a broader context.

Given that I was not one of the previous reviewers, I carefully read all the comments and suggestions made regarding the first version of the manuscript. I agree with the authors' assessment that some of the criticisms seem to stem from a lack of familiarity with macroecological methods or biases against a macroecological approach in studies of sexual selection. While I do not identify myself as a macroecologist, I do appreciate the use of a predictive macroecological approach to understand how sexual selection may operate at large geographic scales.

The criticisms of using a proxy for sexual selection intensity made by reviewers 1 and 2 is valid, but the authors present compelling arguments in favor of the metric used. Additionally, they conduct various analyses to support the validity of this metric and test the robustness of the results in relation to data quality. However, the question remains: despite all the procedures adopted to address the reviewers' criticisms, do the results indeed provide useful information about macroecological gradients in sexual selection intensity? From my perspective, the patterns are solid and the study will undoubtedly not only stimulate discussions but also advance the field of knowledge. Therefore, I believe the work deserves to be published.

Next, I will offer some comments and suggestions on the manuscript. Although I am not an expert in birds and may not feel entirely comfortable evaluating the strength of certain taxon-specific arguments, I hope my input help the authors further improve the quality of the study. If you have any questions, please do not hesitate to contact me.

Glauco Machado

E-mail: glaucom@ib.usp.br

= = = = = = = = = = = = = = = = = = = =

MAJOR COMMENTS

1) The fact that diet modulates the strength of sexual selection in birds is one of the most interesting findings of the study. In the first paragraph of the discussion, the authors suggest that the observed differential response for primary and secondary consumers could explain discrepancies in the reported latitudinal patterns for insectivorous and frugivorous birds. Similarly, the authors propose that discrepancies in latitudinal patterns observed for other taxa could be explained by differences in diet. Unfortunately, I am not certain that the data obtained for birds can shed light on the other taxonomic groups mentioned in the cited articles. Saenz et al. (2006) worked with anurans, which are exclusively secondary consumers; Svensson & Waller (2013) worked with damselflies, also exclusively secondary consumers; Machado et al. (2016) worked with harvestmen, which are omnivores but with a strong tendency towards carnivory; and Krasnov et al. (2022) worked with fleas, which are blood-feeders. In other words, the four mentioned articles deal with taxa that should respond similarly to secondary consumer birds. However, both harvestmen and fleas respond similarly to frugivorous birds from my perspective. Therefore, variations in diet among taxa do not seem to be the primary cause of differences in latitudinal patterns; the explanation may depend on other factors. I think the reproductive biology of birds presents two crucial differences in relation to the taxa mentioned above, hindering a direct comparison of the results: (a) scramble competition is a mating system absent in birds but extremely common in anurans, insects, and harvestmen; and (b) social and genetic monogamy is common in birds but extremely rare in anurans, insects, and harvestmen.

2) "While this may cause a positive link between territoriality and sexual selection in some avian systems, we find that the overall global pattern is reversed, presumably because resource defence polygyny is relatively rare." (L291-293) - OK, but please note that resource defence polygyny is relatively common in damselflies, a taxon in which the latitudinal gradient in the strength of sexual selection is positive (Svensson & Waller 2013), just like birds.

MINOR COMMENTS

L11-12: the expression "mating systems" is already in the title. You could avoid redundancy changing this expression for "migration" or "temperature seasonality".

L61-62: "...either resource defence polygyny or female-only parental care (Machado et al. 2016), leading..."

L72-73: "...as indices of strength of sexual selection..."

L85: "...inferred from mating systems across..."

L126: "...polygamy and EPP (Table 1; see also Methods)."

L139-140: "...Clements et al. (2021) and BirdTree taxomomy..."

L142: "...metrics of sexual selection commonly employed in the literature: residuals..."

L150-151: "which provides a more accurate index of sexual selection (see Discussion)." - I suggest removing this part of the sentence.

L187: "...(negative) in frugivores (Fig. 4c,d)."

L191-192: "...in fruit-eating species (frugivores)..."

L194: Both in the title and on lines 182/202/211/215/217-218/221/241 you refer to "ecology" or "ecological traits" (not life history). Please be consistent throughout the manuscript.

L200-201: "...observed in lower latitudes (Fig. S6)."

L215-216: "(Tables S5-S6)"

L220: "(Fig. 5e; Tables S5-S6)"

L249: "...that the strength of sexual selection..."

L252: "...primary consumers (Fig. 4)."

L266: "...birds was temperatue seasonality..."

L326: I think this section would be better placed after the section "Sexual selection scores for all birds: validity, robustness and research priorities" and before the "Conclusions".

L360: why "sophisticated"?

L423: "...increases from 0 to 4 (Table 1)."

L467: "...mass (Pitcher et al. 2005..."

L566 and 902: et al. - italics

Fig. 2: What do the colors represent in the graphics?

Figs. 4b, 4d, 4h: I suggest limiting the dashed line to the segment of the x-axis for which there is available data.

L1036: "Clements et al. (2021)"

L1069: "...Under Clements taxonomy, P. colchicus..."

L1130-1141: In cases where the Is was separately calculated for males and females of a given species, how was this information used in the database? 

L1276: "...diverse species. Second..."

Figs. S5d, S9b, S9d: I suggest limiting the dashed line to the segment of the x-axis for which there is available data.

Reviewer #2:

** Please note that I have not read the reviews and authors replies for the previous submission of this manuscript to a different journal, that authors have attached with the present manuscript. As I see it, my job as a reviewer is to assess the manuscript submitted to PLoS Biol as it stands, and the manuscript's history of previous reviews and revisions is not relevant. The present ms with its supporting material should stand on its own, and I believe that this alone is what I need to assess.

The ms describes global patterns of geographic variation in the prevalence of sexual selection in birds. This has been done (or attempted) before several times, and in the introduction the authors make a strong case for why this is interesting and important for our understanding of evolution. They do an excellent job of clearly laying out the various alternative hypotheses to explain the possible patterns you could see. This study is probably the most thorough investigation of this issue to date, because it is based on comprehensive global datasets of bird geographic distributions and phylogeny, and the authors have constructed a novel dataset (from secondary data sources) that classifies all bird species into an ordinal-scale classification of sexual selection. They explore the geographic patterns for all birds species and for a number of ecological subsets of birds that allow them to explore nuances in the general pattern, and offer supporting evidence for or against the various mechanistic hypotheses. All in all it is an interesting and rigorous piece of work.

That said, I have a few worries about the degree to which the results support the conclusions the authors draw, which I will list below.

1. Much of the interpretation of latitudinal gradients in SS seems to be based on the p-values of the regression models. This is not ideal as in many cases the huge sample sizes are generating small p-values even when effect sizes are small and the visual pattern (from the maps and the plots) makes a latitudinal trend seem dubious. For example, Figure S3D certainly doesn't look like a latitudinal trend to me, even though the p-value is <0.05 and the authors are happy to take this as support for a positive association holding up for the species with the highest data certainty (line 168-173). I think they just need to be a bit more circumspect with their claims of support for latitudinal trends - use the visuals and effect sizes, not just the p-values, to draw these conclusions.

2. Even with the visual patterns the authors need to be more cautious in their interpretation, or their acceptance of what looks like a trend. For example, in Figure 3C there is a prominent "hotspot" of SS in the Nullarbor Plain / Great Victoria Desert region of south-central Australia, which is a bit odd to say the least, because I'm fairly sure there are few endemic species here, most species that overlap this area are widespread across the southern arid zone, and there's no reason to think the assemblage here should have higher SS on average than surrounding regions. What I think is going on here is that this is an extremely poorly sampled region and this hotspot is likely driven by just one or two species whose ranges overlap the area and which happen to meet the higher data quality and SS criteria. More generally, this makes me wonder how "volatile" these geographic patterns are, ie to what degree the patterns in many parts of the world are driven by small numbers of species. It would be reassuring to know the richness of grid cells for each of these maps. This also makes me question the reliability of some of the analyses based on the aggregate geographic data. 

3. This leads on to an even broader concern. I wonder to what degree the global patterns of SS really are driven by latitude (well, the things that correlate with latitude, obviously), or are simply the results that emerge from the history of radiation of various large bird clades with differing degrees of SS, in different parts of the world - in other words, a pattern driven more by historical contingency than by mechanistic associations with the correlates of latitude. The phylogenetic conservatism of SS (lines 208-209) would seem to support this. I seem to recall this was also the conclusion arrived at by Jetz et al 2012 with respect to global variation in diversification rates. I know the authors have gone some way to addressing this through the inclusion of phylogenetic covariances in their models, but one concern I have is that the models are not spatially explicit (and I don't really see how dichotomizing latitude removes spatial autocorrelation, as suggested in lines 556-558). Phylogenetic and spatial effects are themselves strongly intercorrelated, and really, the only way to handle this in a regression-type model to extract any true underlying geographic pattern is to use methods that incorporate both phylogenetic and spatial covariances. Two examples of how this can be done are (1) Hua et al (2019) Nature Communications 10 (1), 1-10, and (2) Dinnage et al (2020), Proceedings B 287 (1926). The latter paper also addresses another problem the authors mention, which is how to handle species-level and geographic variables in the same model.

4. Line 148-151: I applaud the authors use of these three alternative SS metrics to do a validity check on their index, but generalizing from 14 species to all 10,000 bird species seems to be pushing inductive inference to (beyond?) its limits.

5. Figure 2: what do the three colours represent?

6. Figure 4C: can you really interpret the frugivore pattern as a negative latitudinal gradient, when the data are virtually confined to the tropics?

7. Conclusion: this seems a bit weak, without a clear take-home message, as if it was tacked on at the last minute. I can't offer any specific suggestions but I'd encourage the authors to think a bit more deeply about the general conclusions and broader significance of their findings. I would also remove the bit about relevance to climate change and the future of biodiversity, which seems a bit confected (and unnecessary).

---

## [Decision Letter · Decision Letter 2]

23 Aug 2024

Dear Rob,

Thank you for your patience while we considered your revised manuscript "Climate and ecology predict latitudinal trends in sexual selection inferred from avian mating systems" for publication as a Research Article at PLOS Biology. This revised version of your manuscript has been evaluated by the PLOS Biology editors, the Academic Editor and one of the original reviewers.

Based on the review and our Academic Editor's assessment of your revision, we are likely to accept this manuscript for publication, provided you satisfactorily address the remaining points raised by the reviewer and the following data and other policy-related requests.

IMPORTANT - Please attend to the following:

a) Please could you change your Title very slightly to "Climate and ecology predict latitudinal trends in sexual selection inferred from mating systems in birds"?

b) Please attend to the remaining request from reviewer #2. The Academic Editor notes that you may already have attempted to do this in the previous round, so this is not a hardline requirement, but anything you can do in this direction, bearing in mind our broader readership, would be helpful.

c) Please address my Data Policy requests below; specifically, we need you to supply the numerical values underlying Figs 2ABC, 3ABCDB, 4ABCDEFGH, 5ABCDE, S1ABCD, S2AB, S3ABCD, S4, S5ABCD, S6ABCD, S7ABCDEFGH, S8, S9ABCDEFGH, either as a supplementary data file or as a permanent DOI’d deposition. I note that you already have an associated GitHub deposition (https://www.github.com/ra-barber/sexual_selection), but this is currently not accessible. Because Github depositions can be readily changed or deleted, please make a permanent DOI’d copy (e.g. in Zenodo) and provide this URL (see below); this will need to be made accessible to me (either by setting live or by providing me with a private reviewer link).

d) Please cite the location of the data clearly in all relevant main and supplementary Figure legends, e.g. “The data underlying this Figure can be found in S1 Data” or “The data underlying this Figure can be found in https://zenodo.org/records/XXXXXXXX

e) Please make any custom code available, either as a supplementary file or as part of your data deposition.

We expect to receive your revised manuscript within two weeks. 

*Published Peer Review History*

*Press*

Sincerely,

Roli

Roland Roberts, PhD

Senior Editor

rroberts@plos.org

PLOS Biology

DATA POLICY:

Regardless of the method selected, please ensure that you provide the individual numerical values that underlie the summary data displayed in the following figure panels as they are essential for readers to assess your analysis and to reproduce it: Figs 2ABC, 3ABCDB, 4ABCDEFGH, 5ABCDE, S1ABCD, S2AB, S3ABCD, S4, S5ABCD, S6ABCD, S7ABCDEFGH, S8, S9ABCDEFGH. NOTE: the numerical data provided should include all replicates AND the way in which the plotted mean and errors were derived (it should not present only the mean/average values).

CODE POLICY

DATA NOT SHOWN?

REVIEWER'S COMMENTS:

Reviewer #2:

I have only read briefly through the revised manuscript and the authors replies, but it is clear that authors have genuinely taken on board my original concerns with some aspects of their analyses, and gone to considerable lengths to carry out new analyses of their data to address these. The result is now a far more robust set of results that more strongly support their conclusions. I still think that the conclusion could do with a more clear take-home message or statement of the broader implications of the findings.

---

## [Editor Report · Decision Letter 3]

20 Sep 2024

Dear Joe,

Thank you for the submission of your revised Research Article "Climate and ecology predict latitudinal trends in sexual selection inferred from avian mating systems" for publication in PLOS Biology. On behalf of my colleagues and the Academic Editor, Tiago Quental, I'm pleased to say that we can in principle accept your manuscript for publication, provided you address any remaining formatting and reporting issues. These will be detailed in an email you should receive within 2-3 business days from our colleagues in the journal operations team; no action is required from you until then. Please note that we will not be able to formally accept your manuscript and schedule it for publication until you have completed any requested changes.

Note: I've changed the MS file to indicate that both you and Rob should be flagged as corresponding authors. I've also mentioned it in the notes in our system to the Production dept, for good measure.

Sincerely, 

Roli

Senior Editor

PLOS Biology

rroberts@plos.org